# Axial Length Stabilization or Reduction in over 40% of Patients Wearing Extended Depth-of-Focus Contact Lenses

**DOI:** 10.3390/jcm14051750

**Published:** 2025-03-05

**Authors:** Debabrata Hazra, Erisa Yotsukura, Shihomi Oyama, Yuta Shigeno, Kiwako Mori, Rikako Takeuchi, Yurina Nakajima, Akiko Hanyuda, Mamoru Ogawa, Toshihide Kurihara, Hidemasa Torii, Kazuno Negishi

**Affiliations:** 1Department of Ophthalmology, Keio University School of Medicine, 35 Shinanomachi, Shinjuku-ku, Tokyo 160-8582, Japan; mithunhindu@keio.jp (D.H.); erisa.t@a6.keio.jp (E.Y.); ohym0718@gmail.com (S.O.); yuta0shigeno@yahoo.co.jp (Y.S.); morikiwako@gmail.com (K.M.); takeuchi.rkk@gmail.com (R.T.); yurina00nakajima@gmail.com (Y.N.); akikohanyuda@gmail.com (A.H.); mamo@z5.keio.jp (M.O.); kurihara@z8.keio.jp (T.K.); 2Department of Ophthalmology, Kitasato University Kitasato Institute Hospital, Minato-ku, Tokyo 108-8642, Japan; 3Laboratory of Photobiology, Keio University School of Medicine, 35 Shinanomachi, Shinjuku-ku, Tokyo 160-8582, Japan; 4JINS Endowed Research Laboratory for Myopia, Keio University School of Medicine, Shinjuku-ku, Tokyo 160-8582, Japan

**Keywords:** extended depth-of-focus lenses, myopia, axial length, choroidal thickness, spherical equivalent

## Abstract

**Background/Objectives**: Preceding studies have reported the efficacy of multifocal contact lenses (MFCLs) in slowing myopia progression. Recently, a novel type of MFCL, i.e., extended depth-of-focus (EDOF) contact lens (CL), was designed. Here, we retrospectively investigated myopia progression associated with EDOF CL wear. **Methods**: Twenty-four consecutive myopic children (24 eyes; mean age, 13.9 years) received EDOF CLs to control myopia progression and participated in the study. We measured the axial length (AL), spherical equivalent (SE), and choroidal thickness (CT) at baseline and after 1 year of lens wear and compared the changes. **Results**: The mean baseline AL, SE, and CT were, respectively, 26.31 mm, −6.38 diopter (D), and 235 μm, and at 1 year 26.40 mm (*p* = 0.03), −6.61 D (*p* = 0.05), and 244 μm (*p* = 0.18). The AL decreased in 20.8% of cases (≧−0.05 mm/year), whereas 20.8% and 58.4% of cases had stabilization of the AL or an increased AL (≧+0.05 mm/year), respectively. The patients with a decreased AL engaged in a mean outdoor activity time of 200.6 min/day, the patients with an increased AL (≧+0.05 mm/year) engaged in a mean outdoor activity time of 126.7 min/day. The change in the AL was correlated significantly with the change in the CT (β = −0.46, *p* < 0.05), and 80% of patients with a shortened AL had increased CT (≧+20 μm/year). **Conclusions**: Our data showed that the AL stabilized or decreased in over 40% of myopic patients wearing EDOF CLs.

## 1. Introduction

Myopia, which was estimated to affect about 5 billion people worldwide in 2050 [1], has become a pandemic [2]. We reported previously that both elementary and junior high school students in Tokyo have very high prevalence rates of myopia [3]. We also found that school closures and decreasing time spent outdoors during the COVID-19 pandemic elevated the rate of AL elongation among schoolchildren in Tokyo [4]. The prevalence of high myopia, which elevates the risk of vision loss by retinal detachment, glaucoma-like optic neuropathy, or glaucoma [5] and results in deteriorated mental health of affected individuals [6], also is becoming a serious challenge for the medical community. Therefore, it is crucial to prevent the development of high myopia by suppressing myopia progression [2].

Previous studies have suggested the efficacy of outdoor activities [3,7,8,9,10], low-concentration atropine eye drops, orthokeratology (OK), progressive multifocal spectacle lenses, multifocal contact lenses (MFCLs), including extended depth-of-focus (EDOF) CLs, and low-level red light (LLRL) therapy in myopia control [11,12]. The efficacy of nightly use of 0.05% atropine eyedrops for decreasing the incidence and slowing the progression of myopia was confirmed in a randomized clinical trial [13]. Its efficacy toward slowing myopia progression had a concentration-dependent response [14]. OK’s effect in suppressing myopia progression was proven in a meta-analysis [15]. We showed the short-term effectiveness of VL against myopia progression in a randomized clinical trial [16]. Our group also reported that natural carotenoid crocetin suppressed myopia in a randomized clinical trial [17]. Previous studies reported the efficacy of LLRL in controlling the AL in myopia [18,19], and its effect was associated mainly with the changes in the posterior segment [20]. MFCL can slow myopia progression by from 25% to 72% [12].

However, the efficacy of controlling myopia by using MFCLs varies greatly among studies due to lens design, amount of defocus added to lenses, and participants’ demographics and ethnicities [21,22,23,24,25,26,27,28]. A previous study showed that low-addition soft contact lenses (SCLs) (mipafilcon A; Menicon, Nagoya, Japan) with a decentered optical design reduced the amount of AL elongation in myopic Japanese children [22]. Another study also found that defocus-incorporated SCLs slowed the degree of AL elongation in myopic Hong Kong children [23]. The effect of distance-center bifocal CLs in achieving more than 70% control over myopia progression was confirmed in American children [24]. Chamberlain et al. reported that daily disposable MiSight^®^ (CooperVision, Inc., San Ramon, CA, USA) SCLs produced peripheral myopic defocus that suppressed myopia progression in children in Singapore, Canada, United Kingdom, and Portugal [25]. A meta-analysis confirmed the effect of both concentric ring bifocal and peripheral add multifocal SCLs for myopia control [26]. Recently, a novel type of MFCL, i.e., EDOF CL, that does not provide central or peripheral defocus but provides a better global retinal image quality was designed, and previous studies have confirmed the efficacy of controlling myopia using EDOF CLs [27,28,29]. Here, we report cases with myopia that had stabilized or decreased ALs as the result of wearing EDOF CLs.

## 2. Materials and Methods

### 2.1. Study Design and Study Populations

This retrospective case series adhered to the tenets of the Declaration of Helsinki and was approved by the Keio University School of Medicine Ethics Committee (approval date: 29 October 2018; approval number: 20180189). Since ethical guidelines for clinical studies by the Japanese Ministry of Health, Labor, and Welfare indicate that researchers do not need to obtain written informed consent from each patient for studies not involving biologic tissue but only to review the medical records retrospectively, we displayed our ethical statement and written guidelines for this study on our website in Japanese [30]. Due to the retrospective nature of the study, the Keio University School of Medicine Ethics Committee waived the need for obtaining written informed consent. An opt-out consent method was used to participate in the current research.

Patients prescribed EDOF CLs at the Keio University Hospital from August 2021 to September 2022 were included in the current study. A total of 50 consecutive patients were prescribed EDOF CLs during that period. All participants had a best-corrected visual acuity of >20/20. We excluded those who were on low-concentration atropine eye drops together with EDOF CLs (*n* = 4), had less than a 70% wearing rate of EDOF CLs (*n* = 5), were over 20 years old (*n* = 7), were not prescribed EDOF CLs in the examined eyes due to severe dry eye disease (*n* = 2), and stopped wearing EDOF CLs within a year (*n* = 8). Ultimately, 24 (48.0%) had data available for our study.

### 2.2. Measurements

Patients were prescribed an EDOF CL with a +1.50 D defocus (1 day Pure EDOF Middle; base curve: 8.5 mm; diameter: 14.2 mm, SEED, Tokyo, Japan) that was reported to slow the progression of myopia in a preceding study [27], and patients were followed for a year.

At the first visit and after a year of EDOF CL wear, the participants underwent eye examinations that included measurement of the AL, non-cycloplegic refraction, and CT. Trained orthoptists performed all examinations. The AL was measured by noncontact optical biometry using swept-source optical coherence tomography (OCT) biometry (IOLMaster 700, Carl Zeiss Meditec AG, Jena, Germany), which has a measurement accuracy of ±5 um. We recorded the AL 10 times and averaged the data. The refraction was measured by autorefractometry (TONOREF III, Nidek, Gamagori, Japan). High-resolution OCT (RS-3000 Advance, Nidek, Kyoto, Japan) measured the subfoveal choroidal thickness (CT) between 9:00 a.m. and 3:00 p.m. We defined CT as mentioned previously and measured it using the built-in scale of the OCT instrument.

The patients also completed a lifestyle questionnaire that included factors such as time spent outdoors, performing near work, involvement in outdoor club activities, and parental history of myopia. We defined time spent outdoors as the average number of hours spent outdoors daily that was calculated using the following formula: [(hours spent on a weekday) × 5 + (hours spent on a weekend day) × 2]/7 [31]. Near work included studying; reading books; using a computer, tablet, or smartphone; watching television; and playing video or portable games.

We defined myopia as a spherical equivalent (SE) of ≦−0.5 D in the current study; only the results from the right eye are presented. As IOLMaster 700 (Carl Zeiss Meditec AG) has a reproducibility of 0.02 mm or less [32] and the definition of axial shortening was 0.05 mm in the past studies [19,33,34], we also defined cases in which we observed axial length shortening of 0.05 mm or more as axial length shortening [25,26]. Considering the diurnal variations in the choroid [35], we defined a change in the CT as more than a 20 μm increase or decrease annually.

### 2.3. Statistical Analysis

To evaluate the effect of EDOF CL wear on the AL, SE, and CT, we compared the baseline value and the value after a year of lens wear for each participant using the paired *t*-test. By dividing the patients into the decreased AL group and increased AL group, we compared each factor, including age, baseline AL, baseline SE, baseline CT, time spent outdoors, proportion of outdoor activities time, time spent on near work, and parental history of myopia between the two groups using the Mann–Whitney U-test or Pearson’s chi-square test. We also analyzed the correlation between the change in the AL and the change in the CT by calculating Pearson’s correlation coefficient. Finally, using multiple regression analysis, we examined the associations between the AL and other factors including age, baseline AL, time spent on near work, and time spent outdoors to investigate the factors associated with the change in AL in this study. Using multiple regression analyses, we evaluated the factors via a stepwise analysis.

All statistical analyses were performed using statistical analysis software (SPSS Version 27, International Business Machines Corporation, New York, NY, USA.). All *p* values were considered significant if <0.05.

## 3. Results

### 3.1. Patient Demographics

The mean baseline age of the 24 study patients (boys 54.2%; girls 45.8%) was 13.9 ± 2.8 years (standard deviation [SD]) (range, 8–19 years) (Table 1).

The mean baseline AL, SE, and CT of the participants were 26.31 ± 1.34 mm (range, 23.83–29.32 mm), −6.38 ± 2.47 diopters (D) (−2.01 to −11.62 D), and 235 ± 65 μm (137–391 μm), respectively (Table 2). The mean power of the prescribed EDOF CLs was −5.45 ± 2.07 D (−1.00 to −10.00 D).

### 3.2. Changes in the AL, SE, and CT After Wearing EDOF CLs

The mean AL, SE, and CT of the participants after wearing EDOF CL were 26.40 ± 1.33 mm (24.37–29.48 mm), −6.61 ± 2.48 D (−11.82 to −2.19 D), and 244 ± 72 μm (131–394 μm), respectively (Table 2). The changes in the AL, SE, and CT were 0.09 ± 0.22 mm, −0.23 ± 0.53 D, and 8 ± 32 μm, respectively. The statistical analysis showed a significant change in AL but not in the SE or CT after a year of lens wear (AL: *p* = 0.03; SE: *p* = 0.054; CT: *p* = 0.18) (Table 2). Investigation of the changes in the AL in each case showed that 20.8% of cases had decreased AL (more than a 0.05 mm decrease annually), whereas 20.8% and 58.4% of cases had stabilized the AL or increased the AL, respectively (Figure 1). 

### 3.3. Patient Factors Involved in AL Shortening After Wearing EDOF CLs

We compared each factor between the two groups and found that the decreased-AL group engaged in a mean outdoor activity time of 200.6 min/day, which was longer than that of the increased-AL group who engaged in a mean outdoor activity time of 126.7 min/day, but the difference did not reach significance (Table 3).

### 3.4. Associations Between Changes in AL and Changes in CT After Wearing EDOF CLs

The results showed that the change in the AL correlated significantly with the change in the CT (β = −0.46, *p* < 0.05) (Figure 2), and 80% of cases with a reduced AL had an increased CT (Table 4).

### 3.5. Investigation of Factors Associated with the Changes in AL

Multiple-regression analyses showed that the changes in AL were associated significantly with the baseline age (β = −0.05, *p* = 0.006) (Table 5).

## 4. Discussion

To the best of our knowledge, this study is the first to investigate the changes in the AL, SE, and CT after 1 year of EDOF CL wear in myopic Japanese children. We found that the ALs of 20.8% of cases decreased and 80% of those had increased CT in the current study. Comparison of the patient factors between the shortened AL group and elongated AL group showed that the decreased AL group engaged in longer outdoor activity. We also evaluated the changes in the AL and CT and found that the changes in the AL were correlated significantly with the changes in the CT and that most cases with a decreased AL had an increased CT. Our results suggested that wearing EDOF CL along with outdoor activity lasting longer than 3 h daily may decrease the elongated AL in children with myopia.

Multiple studies have suggested several procedures for controlling myopia progression, such as low-concentration atropine eye drops [13,14], OK [15,36,37,38,39], violet light (VL) therapy [16,40], dietary supplementation of crocetin [17], LLRL [18,19,20,41], and MFCL [21,22,23,24,25,26,27,28]. Of these procedures, an AL decrease in myopic patients was observed using VL therapy [40], OK [37,38,39], and LLRL [18,19,20,41]. Our group reported the case of a 4-year-old boy with anisometropic amblyopia who had decreased AL by using VL transmitting eyeglasses [40]. OK also has been attracting attention widely among myopia specialists, because about 16% of myopic patients using OK experienced AL decreases, and the incidence rate increased with the baseline age [38]. Wang et al. found that more than 20% of children receiving LLRL therapy achieved AL decreases [18,19]. In the current study, we found decreased ALs in myopic children as a result of EDOF CL wear, a type of MFCL. Considering that optical treatments such as OK and MFCL seem to produce less of a rebound effect [11], we suggest that MFCL may be an effective and less risky treatment for not only suppressing myopia progression but also decreasing the degree of already developed myopia.

To date, many studies have investigated the efficacy of wearing MFCLs for controlling myopia progression [21,22,23,24,25,26,27,28]. A previous study indicated that dual-focal soft CLs decreased AL elongation by producing myopic defocus without compromising visual function in myopic children in New Zealand [21]. Chamberlain et al. reported that daily disposable MiSight soft CLs that produce peripheral myopic defocus suppressed myopia progression in children in Singapore, Canada, United Kingdom, and Portugal [25]. A meta-analysis was conducted and confirmed the effect of both concentric ring bifocal and peripheral add multifocal soft CLs for myopia control [26]. Recently, a novel type of MFCL, i.e., EDOF CL, that does not provide central or peripheral defocus but provides a better global retinal image quality was designed, and its efficacy for suppressing myopia progression was reported in Chinese (10.5 ± 1.3 years) [27], Taiwanese (12.4 ± 1.5 years) [28], and Indian (11.3 ± 2.6 years) [29] children. In this study of myopic Japanese children, we found that more than 20% of cases had a decreased AL resulting from wearing EDOF CLs. The amount of AL changes was smaller in the current study (0.09 ± 0.19 mm) compared to that of previous studies of Chinese (0.22 ± 0.16 mm) [27], Taiwanese (0.34 ± 0.19 mm) [28], and Indian (0.11 ± 0.03 mm) [29] children who wore EDOF CLs. This may be attributed to the higher baseline age in the current study (13.9 ± 2.8 years), as the change in the AL was associated significantly with the baseline age (Table 5). The advantages of EDOF CLs are that they are disposable and safe, and they are compatible with high myopia. However, they have limitations, in that they cannot correct astigmatism, and that patients must wear CLs during the day.

Although it is thought that they may reduce hyperopic defocus in the peripheral retina [21], which may modulate the choroidal blood flow to change the tissue’s thickness [42,43,44,45], there is still no consensus as to how EDOF CLs slow myopia progression in schoolchildren. The choroid plays a crucial role in the modulation of AL [46], and its thickness is correlated negatively with the AL [47,48]. In the current study, we observed choroidal thickening as a result of EDOF CL wear, so we believe that choroidal thickening may also be one of the mechanisms of suppression of myopia progression by EDOF CLs.

To the best of our knowledge, this is the first study to report cases of decreased ALs as a result of wearing EDOF CLs in myopic children and to investigate the efficacy of EDOF CLs in controlling myopia progression in a Japanese population. To investigate the factors that affect the efficiency of EDOF CLs in myopia control, we compared each factor between the group with a decreased AL and the group with an increased AL and found that the former was slightly older at baseline and engaged in slightly longer outdoor activity time of more than 3 h daily on average (Table 3). Since outdoor activities effectively suppress myopia progression [3,7,8,9,10], longer outdoor activity time might have contributed to controlling the myopia in the current study. The biological mechanism explaining how outdoor activities suppress myopia progression is not fully understood, although the role of VL in association with outdoor light environment has been suggested in our previous studies [49,50]. We showed the suppression rates for AL elongation by both VL-transmitting CLs [49] and glasses [50]. Our group also reported that the VL effect was dependent on retinal expression of the VL-sensitive atypical opsin, neuropsin (Opn5) [51]. With outdoor activity exceeding 3 h daily, our results indicated that EDOF CLs can not only control myopia progression but also reduce the elongated AL of myopia in Japanese children. However, multiple regression analyses showed that the change in the AL was associated significantly with the baseline age but not with outdoor activity time (Table 5). This might be attributed to the long outdoor activity time of the total study population for more than 2 h daily on average and to the potential confounding factors like near work and screen exposure.

The current study had some limitations. First, because this study did not include a control group, we could not fully isolate the effects of the EDOF CLs in controlling myopia. Second, the sample size was small, and because of that, we could not include factors other than outdoor activity, such as genetic factors and near work time, in our multiple-factor analysis. Third, one orthoptist measured the CT manually. Finally, selection bias was possible because the study included only those who visited the Keio University Hospital Myopia Clinic.

In conclusion, we observed 20.8% of cases with decrease in the ALs in eyes wearing EDOF CLs, and 80% of those had increased CT in the current study. Because the cases with a decreased AL engaged in longer higher outdoor activity of more than 3 h daily on average compared with the cases with increased ALs, we suggest that wearing EDOF CLs with sufficient outdoor activity time may reduce the elongated AL of myopia.

## Figures and Tables

**Figure 1 jcm-14-01750-f001:**
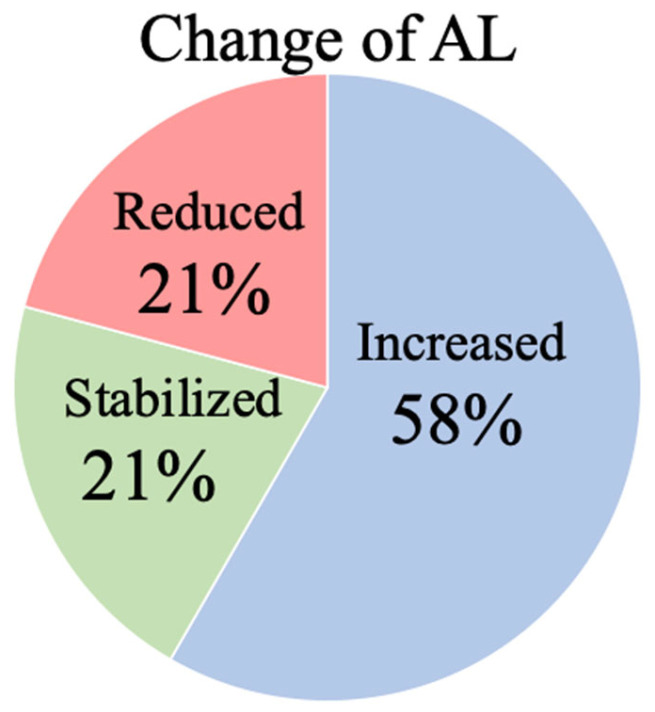
The proportions of patients in the increased-AL group, stabilized-AL group, and reduced-AL group after a year of EDOF CL wear: 58.4% of cases had an increased AL (≧+0.05 mm/year), 20.8% of cases had a stabilized AL, and 20.8% of cases had a decreased AL (≧−0.05 mm/year). The AL stabilized or reduced in over 40% of patients wearing EDOF CLs.

**Figure 2 jcm-14-01750-f002:**
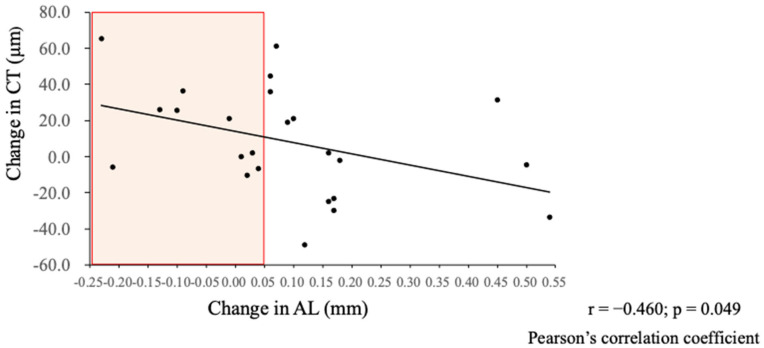
The correlations between the change in the AL and the change in the CT after a year of wearing EDOF CLs. Pearson’s correlation coefficient and *p* value are shown between the change in the AL and the change in the CT. Significant correlations are seen between the change in the AL and the change in the CT (*β* = −0.46, *p* < 0.05). The highlighted area indicates the cases in which the AL stabilized or decreased.

**Table 1 jcm-14-01750-t001:** Subject demographic data.

Parameter	Mean ± SD (Range)
Gender	Male 13 (54.2%); female 11 (45.8%)
Age (years)	13.9 ± 2.8 (8–19)
Prescribed CL (D)	−5.45 ± 2.07 (−1.00 to 10.00)
Outdoor activity time (minutes)	146.0 ± 80.8 (30–369)
Involvement in outdoor club activities	33.3%
Near work time (minutes)	93.0 ± 43.5 (20–174)
Parental history of myopia (either or both)	95.8%

Data are expressed as the mean ± standard deviation.

**Table 2 jcm-14-01750-t002:** Changes in the non-cycloplegic objective refraction, AL, and CT with EDOF CL wear for 1 year.

Parameters	Baseline Mean ± SD	After Wearing EDOF CLs Mean ± SD	Change Mean ± SD	*p* Value
Non-cycloplegic objective refraction (D)	−6.38 ± 2.47	−6.61 ± 2.48	−0.23 ± 0.53	0.054
AL (mm)	26.31 ± 1.34	26.40 ± 1.33	0.09 ± 0.22	**0.03**
CT (μm)	235 ± 65	244 ± 72	8 ± 32	0.18

Data are expressed as the mean ± standard deviation. Baseline value and the value after a year of EDOF lens wear for each participant were compared using the paired *t*-test. *p* values < 0.05 are bold.

**Table 3 jcm-14-01750-t003:** Comparison of factors between axial length elongated group and shortened group with EDOF CL wearing for 1 year.

Parameters	Increased Group (*n* = 14); Mean ± SD	Decreased Group (*n* = 5); Mean ± SD	*p* Value
Age at Prescription (year)	12.9 ± 2.9	15.4 ± 2.1	0.14
Objective Refraction at Prescription (diopter)	−6.65 ± 2.77	−6.96 ± 2.16	0.71
Axial Length at Prescription (mm)	26.37 ± 1.35	26.88 ± 1.32	0.58
Choroidal Thickness at Prescription (μm)	216.68 ± 53.39	202.60 ± 67.87	0.46
Outdoor Activities Time (minute)	126.7 ± 62.6	200.6 ± 92.3	0.13
Involvement in Outdoor Club Activities	42.9%	40.0%	0.66
Near Work Time (minute)	86.0 ± 46.4	111.6 ± 18.0	0.21
Parental History of Myopia (either or both)	100.0%	80.0%	0.26

Data are expressed as the mean ± SD. SD = standard deviation. Values were compared using the Mann–Whitney U test or Pearson’s chi-square test.

**Table 4 jcm-14-01750-t004:** Change in axial length and change in choroidal thickness in axial length shortened cases with EDOF CL wearing for 1 year.

Cases	Changes in Axial Length (mm/Year)	Changes in Choroidal Thickness (μm/Year)
1	−0.09	+37
2	−0.10	+26
3	−0.13	+26
4	−0.21	−6
5	−0.23	+66

Defining change in the axial length and the choroidal thickness as more than 0.05 mm and 20 μm increase or decrease per year, respectively, we found 80.0% of cases with decreased AL had increased CT.

**Table 5 jcm-14-01750-t005:** Results of multiple-regression analyses to investigate the factors associated with the change in axial length.

	Change in AL, (mm) (*n* = 24)		
	β	95% CI	*p* Value
Age (years)	−0.05	−0.08 to −0.02	**0.006**
Baseline AL (mm)	−0.02	−0.08 to 0.05	0.59
Near work time (minutes)	−0.00056	−0.0033 to 0.0021	0.66
Outdoor activity time (minutes)	0.00017	−0.0013 to 0.0016	0.80
R^2^		0.48	

A multiple-regression model was used. The change in the AL was used as the outcome variable. *p* values < 0.05 are highlighted. *β* = coefficient; 95% CI = 95% confidence interval.

## Data Availability

D.H. and Y.S. had full access to all the study data and take responsibility for the integrity of the data and the accuracy of the data analysis. The data that support the findings of this study are available from D.H. and Y.S., but restrictions apply to the availability of these data, which were used under license for the current study, and so are not publicly available. However, data are available from the authors upon reasonable request and with the permission of H.T.

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
