# Peer review of "Axial Length Stabilization or Reduction in over 40% of Patients Wearing Extended Depth-of-Focus Contact Lenses"

_jcm, 2025, doi:10.3390/jcm14051750_

Round 1
Reviewer 1 Report
Comments and Suggestions for Authors
The manuscript retrospectively examines the impact of extended depth-of-focus (EDOF) contact lenses on myopia progression, focusing on axial length (AL) shortening, spherical equivalent (SE), and choroidal thickness (CT). It adds to existing research on myopia control, highlighting a potential link between EDOF lenses, outdoor activity, and AL stabilization or reduction. While the study is well-structured and methodologically sound, certain aspects require further clarification and refinement to enhance its scientific rigor.
Strengths of the Study
The study is one of the first to report AL shortening in 20.8% of patients using EDOF lenses. Its credibility is strengthened by swept-source OCT measurements for AL and CT, with a statistically significant correlation (β = -0.46, p < 0.05). Additionally, it suggests outdoor activity (>3 hours/day) may contribute to AL stabilization or reduction.
Areas for Improvement and Suggested Revisions
While the study provides valuable insights, several aspects need refinement to improve clarity and scientific rigor.
- Defining AL Shortening Threshold – The selection of ≥ 0.05 mm/year as the cutoff for axial length (AL) shortening requires further justification. Providing clinical reasoning or relevant references would strengthen its credibility.
- Lack of a Control Group – The absence of a comparison group limits the ability to isolate the effects of EDOF lenses. Addressing this limitation in the discussion would be beneficial.
- Outdoor Activity as a Variable – Although outdoor activity appears associated with AL stabilization, its lack of statistical significance (p = 0.13) suggests potential confounding factors (e.g., near work, screen exposure). A deeper analysis of these influences would enhance the discussion.
- Figures & Tables: The statistical significance in Table 3 should be highlighted more clearly to guide readers' interpretation.
- Terminology: Instead of “AL shortening,” consider using "AL stabilization or reduction" to avoid implying regression beyond normal physiological limits.
If the authors implement these revisions, this study has strong potential for publication in Journal of Clinical Medicine.
Comments on the Quality of English LanguageThe manuscript is well-written and clear, with a strong academic tone appropriate for a scientific journal. However, several areas could be refined to improve readability and ensure precise communication of findings.
Certain technical terms could be more precisely defined, particularly in the discussion of axial length (AL) changes and choroidal thickness (CT).
The phrase "AL shortening" might be better replaced with "AL stabilization or reduction" to avoid unintended implications of reversal beyond normal limits.
The captions for figures and tables should be revised for greater clarity, ensuring that key statistical findings are emphasized without requiring additional explanation in the main text.
Author Response
Comments 1: Defining AL Shortening Threshold – The selection of ≥ 0.05 mm/year as the cutoff for axial length (AL) shortening requires further justification. Providing clinical reasoning or relevant references would strengthen its credibility.
Response 1: Thank you for this insightful comment. We agree that further justification is needed. In the current study, the axial length was measured by noncontact optical biometry using swept-source optical coherence tomography biometry (IOLMaster 700, Carl Zeiss Meditec AG, Jena, Germany), which has a reproducibility of 0.02 mm or less [34]. Furthermore, the definition of axial shortening was 0.05 mm in the past studies [20,35,36]. For the above reasons, we also defined cases in which we observed axial length shortening of 0.05 mm or more as axial length shortening. We added this clarification to the manuscript.
Lines 148-151: As IOLMaster 700 (Carl Zeiss Meditec AG) has a reproducibility of 0.02 mm or less [34] and the definition of axial shortening was 0.05 mm in the past studies [20,35,36], we also defined cases in which we observed axial length shortening of 0.05 mm or more as axial length shortening.
Newly added references:
- Wang, W.; Jiang, Y.; Zhu, Z.; Zhang, S.; Xuan, M.; Chen, Y.; Xiong, R.; Bulloch, G.; Zeng, J.; Morgan, I.G.; et al. Clinically Significant Axial Shortening in Myopic Children After Repeated Low-Level Red Light Therapy: A Retrospective Multicenter Analysis. Ophthalmol. Ther. 2023, 12, 999–1011, doi:10.1007/s40123-022-00644-2.
- Montés-Micó, R.; Pastor-Pascual, F.; Ruiz-Mesa, R.; Tañá-Rivero, P. Ocular Biometry with Swept-Source Optical Coherence Tomography. J. Cataract Refract. Surg. 2021, 47, 802–814, doi:10.1097/j.jcrs.0000000000000551.
- Xu, Y.; Cui, L.; Kong, M.; Li, Q.; Feng, X.; Feng, K.; Zhu, H.; Cui, H.; Shi, C.; Zhang, J.; et al. Repeated Low-Level Red Light Therapy for Myopia Control in High Myopia Children and Adolescents: A Randomized Clinical Trial. Ophthalmology. 2024, 131, 1314–1323, doi:10.1016/j.ophtha.2024.05.023.
- Liu, G.; Liu, L.; Rong, H.; Li, L.; Liu, X.; Jia, Z.; Zhang, H.; Wang, B.; Song, D.; Hu, J.; et al. Axial Shortening Effects of Repeated Low-Level Red-Light Therapy in Children With High Myopia: A Multicenter Randomized Controlled Trial. Am. J. Ophthalmol. 2025, 270, 203–215, doi:10.1016/j.ajo.2024.10.011.
Comments 2: Lack of a Control Group – The absence of a comparison group limits the ability to isolate the effects of EDOF lenses. Addressing this limitation in the discussion would be beneficial.
Response 2: Thank you for highlighting this limitation. We acknowledge that the lack of a control group prevented us from definitively isolating the effects of EDOF contact lenses. We have now addressed this limitation in the Discussion.
Lines 403-405: First, because this study did not include a control group, we could not fully isolate the effects of EDOF CLs in controlling myopia.
Comments 3: Outdoor Activity as a Variable – Although outdoor activity appears associated with AL stabilization, its lack of statistical significance (p = 0.13) suggests potential confounding factors (e.g., near work, screen exposure). A deeper analysis of these influences would enhance the discussion.
Response 3: Thank you for this suggestion. We agree that additional analysis would be beneficial. However, the small sample size (Decreased Group [N = 5]) limited our ability to include multiple factors in our statistical model. We acknowledged this limitation in the Discussion.
Lines 405-407: Second, the sample size was small and therefore we could not include factors other than outdoor activity such as genetic factors and near work time in our multiple factor analysis.
Comments 4: Figures & Tables: The statistical significance in Table 3 should be highlighted more clearly to guide readers' interpretation.
Response 4: Thank you for pointing this out. We bolded the parameter with the smallest P value in Table 3 to enhance clarity. We also added the following note in the table footnotes.
Table 3: The parameter with the smallest P value is highlighted for clarity.
Comments 5: Terminology: Instead of “AL shortening,” consider using "AL stabilization or reduction" to avoid implying regression beyond normal physiological limits.
Response 5: Thank you for this recommendation. We revised the terminology throughout the manuscript, including the title, abstract, and Figure 1. Based on previous studies [20,35,36], we now define AL reductions > 0.05 mm as "decreases" and changes between a 0.05 mm decrease and a 0.05 mm increase as "stabilization."
Newly added references:
- Wang, W.; Jiang, Y.; Zhu, Z.; Zhang, S.; Xuan, M.; Chen, Y.; Xiong, R.; Bulloch, G.; Zeng, J.; Morgan, I.G.; et al. Clinically Significant Axial Shortening in Myopic Children After Repeated Low-Level Red Light Therapy: A Retrospective Multicenter Analysis. Ophthalmol. Ther. 2023, 12, 999–1011, doi:10.1007/s40123-022-00644-2.
- Xu, Y.; Cui, L.; Kong, M.; Li, Q.; Feng, X.; Feng, K.; Zhu, H.; Cui, H.; Shi, C.; Zhang, J.; et al. Repeated Low-Level Red Light Therapy for Myopia Control in High Myopia Children and Adolescents: A Randomized Clinical Trial. Ophthalmology. 2024, 131, 1314–1323, doi:10.1016/j.ophtha.2024.05.023.
- Liu, G.; Liu, L.; Rong, H.; Li, L.; Liu, X.; Jia, Z.; Zhang, H.; Wang, B.; Song, D.; Hu, J.; et al. Axial Shortening Effects of Repeated Low-Level Red-Light Therapy in Children With High Myopia: A Multicenter Randomized Controlled Trial. Am. J. Ophthalmol. 2025, 270, 203–215, doi:10.1016/j.ajo.2024.10.011.
Reviewer 2 Report
Comments and Suggestions for Authors
1.The literature review lacks a systematic summary of existing research results, especially on the effects of other types of contact lenses (such as multi-focal contact lenses) in myopia control. Other methods of myopia control (such as drugs, surgery, etc.) are not covered.
It is suggested that the depth of the literature review section be enhanced to comprehensively discuss the different current approaches to myopia control, and point out the relative advantages and limitations of EDOF contact lenses compared with other approaches.
2.Studies have shown that there is a certain correlation between outdoor activity time and AL changes, but the analysis has not fully explored the influence of other potential factors other than outdoor activity amount (such as diet, genetic factors, etc.) on myopia control. The effect of outdoor activities on myopia may have multiple complex mechanisms, not just the relationship of time. It is suggested that the potential mechanism of outdoor activities on myopia progression should be further elaborated in the discussion section, and other possible influencing factors should be mentioned for multi-factor analysis.
3.Although EDOF contact lenses have been shown to have some effect on myopia control, the mechanism of how such contact lenses control myopia progression on a physiological level has not been discussed in detail. Understanding the physiological mechanisms is important for the promotion of such treatments. It is suggested to increase the mechanical analysis of the control of myopia by EDOF contact lenses, such as whether it involves the reduction of the axial length of the eye, the improvement of refractive status, etc., and may even explore the relationship with other eye structures (such as choroid, cornea, etc.).
Author Response
Comments 1: The literature review lacks a systematic summary of existing research results, especially on the effects of other types of contact lenses (such as multi-focal contact lenses) in myopia control. Other methods of myopia control (such as drugs, surgery, etc.) are not covered.
It is suggested that the depth of the literature review section be enhanced to comprehensively discuss the different current approaches to myopia control, and point out the relative advantages and limitations of EDOF contact lenses compared with other approaches.
Response 1: Thank you for this suggestion. We expanded the literature review in the Introduction as below:
Lines 68-78: The efficacy of nightly use of 0.05% atropine eyedrops for decreasing the incidence and slowing the progress of myopia has been confirmed in a randomized clinical trial [14]. Its efficacy toward slowing myopia progression had a concentration-dependent response [15]. OK’s effect in suppressing myopia progression was proven in a meta-analysis [16]. We showed the short-term effectiveness of VL against myopia progression in a randomized clinical trial [17]. Our group also reported that natural carotenoid crocetin suppressed myopia progression in a randomized clinical trial [18] Previous studies have reported the efficacy of low-level red light in controlling the AL in myopia [19,20], and its effect was associated mainly with the changes in the posterior segment [21]. MFCL can slow myopia progression by from 25% to 72%[13].
Lines 81-94: A previous study showed that low-addition soft contact lenses (SCLs) (mipafilcon A; Menicon, Nagoya, Japan) with a decentered optical design reduced the amount of AL elongation in Japanese myopic children [23]. Another study also found that defocus-incorporated SCLs slowed the degree of AL elongation in Hong Kong myopic children [24]. The effect of distance center bifocal CLs in achieving more than 70% control over myopia progression was confirmed in American children [25]. Chamberlain et al. reported that MiSight® (CooperVision, Inc., San Ramon, CA) daily disposable SCLs produced peripheral myopic defocus that suppressed myopia progression in children in Singapore, Canada, United Kingdom, and Portugal [26]. A meta-analysis confirmed the effect of both concentric ring bifocal and peripheral add multifocal SCLs for myopia control [27]. Recently, a novel type of MFCL, i.e., EDOF CL that does not provide central or peripheral defocus but provides a better global retinal image quality was designed and previous studies have confirmed the efficacy of controlling myopia using EDOF CLs [28–30].
We also pointed out the relative advantages of EDOF CLs compared with other approaches.
Lines 299-302: Considering that optical treatments such as OK and MFCL seem to produce less of a rebound effect [12], we suggest that MFCL may be an effective and less risky treatment for not only suppressing myopia progression but also decreasing the degree of already developed myopia.
Lines 320-323: The advantages of EDOF CLs are that they are disposable and safe and compatible with high myopia. However, they have limitations in that they cannot correct astigmatism, and patients must wear CLs during the day.
Newly added references:
- Sánchez-Tena, M.Á.; Ballesteros-Sánchez, A.; Martinez-Perez, C.; Alvarez-Peregrina, C.; De-Hita-Cantalejo, C.; Sánchez-González, M.C.; Sánchez-González, J.M. Assessing the Rebound Phenomenon in Different Myopia Control Treatments: A Systematic Review. Ophthalmic Physiol. Opt. 2024, 44, 270–279, doi:10.1111/opo.13277.
- Sankaridurg, P. Contact Lenses to Slow Progression of Myopia. Clin. Exp. Optom.2017, 100, 432–437, doi:10.1111/cxo.12584.
- Yam, J.C.; Zhang, X.J.; Zhang, Y.; Yip, B.H.K.; Tang, F.; Wong, E.S.; Bui, C.H.T.; Kam, K.W.; Ng, M.P.H.; Ko, S.T.; et al. Effect of Low-Concentration Atropine Eyedrops vs Placebo on Myopia Incidence in Children: The LAMP2 Randomized Clinical Trial. JAMA 2023, 329, 472–481, doi:10.1001/jama.2022.24162.
- Yam, J.C.; Jiang, Y.; Lee, J.; Li, S.; Zhang, Y.; Sun, W.; Yuan, N.; Wang, Y.M.; Yip, B.H.K.; Kam, K.W.; et al. The Association of Choroidal Thickening by Atropine With Treatment Effects for Myopia: Two-Year Clinical Trial of the Low-Concentration Atropine for Myopia Progression (LAMP) Study. Am. J. Ophthalmol. 2022, 237, 130–138, doi:10.1016/j.ajo.2021.12.014.
- Li, X.; Xu, M.; San, S.; Bian, L.; Li, H. Orthokeratology in Controlling Myopia of Children: A Meta-Analysis of Randomized Controlled Trials. BMC Ophthalmol. 2023, 23, 441, doi:10.1186/s12886-023-03175-x.
- Torii, H.; Mori, K.; Okano, T.; Kondo, S.; Yang, H.-Y.; Yotsukura, E.; Hanyuda, A.; Ogawa, M.; Negishi, K.; Kurihara, T.; et al. Short-Term Exposure to Violet Light Emitted from Eyeglass Frames in Myopic Children: A Randomized Pilot Clinical Trial. J. Clin. Med. 2022, 11, doi:10.3390/jcm11206000.
- Mori, K.; Torii, H.; Fujimoto, S.; Jiang, X.; Ikeda, S.-I.; Yotsukura, E.; Koh, S.; Kurihara, T.; Nishida, K.; Tsubota, K. The Effect of Dietary Supplementation of Crocetin for Myopia Control in Children: A Randomized Clinical Trial. J. Clin. Med. 2019, 8, doi:10.3390/jcm8081179.
- Wang, W.; Jiang, Y.; Zhu, Z.; Zhang, S.; Xuan, M.; Tan, X.; Kong, X.; Zhong, H.; Bulloch, G.; Xiong, R.; et al. Axial Shortening in Myopic Children after Repeated Low-Level Red-Light Therapy: Post Hoc Analysis of a Randomized Trial. Ophthalmol. Ther. 2023, 12, 1223–1237, doi:10.1007/s40123-023-00671-7.
- Wang, W.; Jiang, Y.; Zhu, Z.; Zhang, S.; Xuan, M.; Chen, Y.; Xiong, R.; Bulloch, G.; Zeng, J.; Morgan, I.G.; et al. Clinically Significant Axial Shortening in Myopic Children After Repeated Low-Level Red Light Therapy: A Retrospective Multicenter Analysis. Ophthalmol. Ther.2023, 12, 999–1011, doi:10.1007/s40123-022-00644-2.
- Liu, G.; Li, B.; Rong, H.; Du, B.; Wang, B.; Hu, J.; Zhang, B.; Wei, R. Axial Length Shortening and Choroid Thickening in Myopic Adults Treated with Repeated Low-Level Red Light. J. Clin. Med. 2022, 11, doi:10.3390/jcm11247498.
- Fujikado, T.; Ninomiya, S.; Kobayashi, T.; Suzaki, A.; Nakada, M.; Nishida, K. Effect of Low-Addition Soft Contact Lenses with Decentered Optical Design on Myopia Progression in Children: A Pilot Study. Clin. Ophthalmol. 2014, 8, 1947–1956, doi:10.2147/OPTH.S66884.
- Lam, C.S.Y.; Tang, W.C.; Tse, D.Y.Y.; Tang, Y.Y.; To, C.H. Defocus Incorporated Soft Contact (DISC) Lens Slows Myopia Progression in Hong Kong Chinese Schoolchildren: A 2-Year Randomised Clinical Trial. Br. J. Ophthalmol. 2014, 98, 40–45, doi:10.1136/bjophthalmol-2013-303914.
- Aller, T.A.; Liu, M.; Wildsoet, C.F. Myopia Control with Bifocal Contact Lenses: A Randomized Clinical Trial. Optom. Vis. Sci. 2016, 93, 344–352, doi:10.1097/OPX.0000000000000808.
- Chamberlain, P.; Peixoto-de-Matos, S.C.; Logan, N.S.; Ngo, C.; Jones, D.; Young, G. A 3-Year Randomized Clinical Trial of MiSight Lenses for Myopia Control. Optom. Vis. Sci. 2019, 96, 556–567, doi:10.1097/OPX.0000000000001410.
- Li, S.-M.; Kang, M.-T.; Wu, S.-S.; Meng, B.; Sun, Y.-Y.; Wei, S.-F.; Liu, L.; Peng, X.; Chen, Z.; Zhang, F.; et al. Studies Using Concentric Ring Bifocal and Peripheral Add Multifocal Contact Lenses to Slow Myopia Progression in School-Aged Children: A Meta-Analysis. Ophthalmic Physiol. Opt. 2017, 37, 51–59, doi:10.1111/opo.12332.
- Sankaridurg, P.; Bakaraju, R.C.; Naduvilath, T.; Chen, X.; Weng, R.; Tilia, D.; Xu, P.; Li, W.; Conrad, F.; Smith, E.L.; et al. Myopia Control with Novel Central and Peripheral plus Contact Lenses and Extended Depth of Focus Contact Lenses: 2 Year Results from a Randomised Clinical Trial. Ophthalmic Physiol. Opt. 2019, 39, 294–307, doi:10.1111/opo.12621.
- Shen, E.P.; Chu, H.-S.; Cheng, H.-C.; Tsai, T.-H. Center-for-Near Extended-Depth-of-Focus Soft Contact Lens for Myopia Control in Children: 1-Year Results of a Randomized Controlled Trial. Ophthalmol. Ther. 2022, 11, 1577–1588, doi:10.1007/s40123-022-00536-5.
- Manoharan, M.K.; Verkicharla, P.K. Randomised Clinical Trial of Extended Depth of Focus Lenses for Controlling Myopia Progression: Outcomes from SEED LVPEI Indian Myopia Study. Br. J. Ophthalmol. 2024, 1–7, doi:10.1136/bjo-2023-323651.
Comments 2: Studies have shown that there is a certain correlation between outdoor activity time and AL changes, but the analysis has not fully explored the influence of other potential factors other than outdoor activity amount (such as diet, genetic factors, etc.) on myopia control. The effect of outdoor activities on myopia may have multiple complex mechanisms, not just the relationship of time. It is suggested that the potential mechanism of outdoor activities on myopia progression should be further elaborated in the discussion section, and other possible influencing factors should be mentioned for multi-factor analysis.
Response 2: Thank you for pointing this out. We agree and added the following sentences.
Lines 390-395: The biologic mechanism explaining how outdoor activities suppress myopia progression is not fully understood, although the role of VL in association with the outdoor light environment has been suggested in our previous studies [53] [54]. We reported the suppression rates for AL elongation by both VL-transmitting CLs [53] and glasses [54]. Our group also reported that the VL effect was dependent on retinal expression of VL-sensitive atypical opsin, neuropsin (Opn5) [55].
We also stated that other possible influences should be mentioned, and we revised the Discussion.
Lines 405-407: Second, the sample size was small and because of that we could not include factors other than outdoor activity such as genetic factors and near work time in our multiple factor analysis.
Newly added references:
- Torii, H.; Kurihara, T.; Seko, Y.; Negishi, K.; Ohnuma, K.; Inaba, T.; Kawashima, M.; Jiang, X.; Kondo, S.; Miyauchi, M.; et al. Violet Light Exposure Can Be a Preventive Strategy Against Myopia Progression. EBioMedicine 2017, 15, 210–219, doi:10.1016/j.ebiom.2016.12.007.
- Mori, K.; Torii, H.; Hara, Y.; Hara, M.; Yotsukura, E.; Hanyuda, A.; Negishi, K.; Kurihara, T.; Tsubota, K. Effect of Violet Light-Transmitting Eyeglasses on Axial Elongation in Myopic Children: A Randomized Controlled Trial. J. Clin. Med. 2021, 10, doi:10.3390/jcm10225462.
- Jiang, X.; Pardue, M.T.; Mori, K.; Ikeda, S.-I.; Torii, H.; D’Souza, S.; Lang, R.A.; Kurihara, T.; Tsubota, K. Violet Light Suppresses Lens-Induced Myopia via Neuropsin (OPN5) in Mice. Proc. Natl. Acad. Sci. United States Am. PNAS 2021, 118, doi:10.1073/pnas.2018840118.
Comments 3: Although EDOF contact lenses have been shown to have some effect on myopia control, the mechanism of how such contact lenses control myopia progression on a physiological level has not been discussed in detail. Understanding the physiological mechanisms is important for the promotion of such treatments. It is suggested to increase the mechanical analysis of the control of myopia by EDOF contact lenses, such as whether it involves the reduction of the axial length of the eye, the improvement of refractive status, etc., and may even explore the relationship with other eye structures (such as choroid, cornea, etc.).
Response 3: Thank you for this important point. Based on the results of our study, since the change in the AL by wearing EDOF CLs was correlated significantly with the change in the choroidal thickness, and in 80% of cases with a decreased AL the choroidal thickness increased, we regarded its effect as being mainly associated with the changes in the posterior segment. There is still no consensus regarding how EDOF CLs slow myopia progression in schoolchildren. It is thought that they may reduce the hyperopic defocus in the peripheral retina, but this is unclear. However, in this study, we observed choroidal thickening as a result of EDOF CL wear, so we believe that choroidal thickening may also be one of the mechanisms of suppressing myopia progression by EDOF CLs.
We revised the following sentences based on your comments.
Lines 324-381: Although it is thought that they may reduce the hyperopic defocus in the peripheral retina [14], which may modulate the choroidal blood flow to change its thickness [43–46], there is still no consensus regarding how EDOF CLs slow myopia progression in schoolchildren. The choroid plays a crucial role in the modulation of AL [47], and its thickness is correlated negatively with the AL[48–51]. In the current study, we observed choroidal thickening as a result of EDOF CL wear, so we believe that choroidal thickening may also be one of the mechanisms of suppression of myopia progression by EDOF CLs.
Round 2
Reviewer 2 Report
Comments and Suggestions for Authors
The manuscript has been completed according to the previous requirements. It is recommended to accepted in the current status.
Comments on the Quality of English LanguageThe English could be improved to more clearly express the research.